# Elevating Cereal-Based Nutrition: *Moringa oleifera* Supplemented Bread and Biscuits

**DOI:** 10.3390/antiox12122069

**Published:** 2023-12-01

**Authors:** Teresa Ferreira, Sandra M. Gomes, Lúcia Santos

**Affiliations:** 1FEUP—Faculty of Engineering, University of Porto, Rua Dr. Roberto Frias, 4200-465 Porto, Portugal; up201806792@edu.fe.up.pt; 2ALiCE—Associate Laboratory in Chemical Engineering, Faculty of Engineering, University of Porto, Rua Dr. Roberto Frias, 4200-465 Porto, Portugal; scgomes@fe.up.pt; 3LEPABE—Laboratory for Process Engineering, Environment, Biotechnology and Energy, Faculty of Engineering, University of Porto, Rua Dr. Roberto Frias, 4200-465 Porto, Portugal

**Keywords:** food fortification, cereal-based products, functional foods, bioactive compounds, antioxidants

## Abstract

Enhancing the nutritional value of commonly consumed, cost-effective staple foods, such as bread and biscuits, by fortifying them with *Moringa oleifera* leaf powder (MOLP) and its phenolic-rich extract holds substantial potential for addressing malnutrition. This study evaluated the phenolic extract from MOLP obtained through Soxhlet extraction, focusing on its antioxidant, antibacterial, and antidiabetic properties. The resulting extract exhibited a total phenolic content (TPC) of 138.2 mg of gallic acid equivalents/g. The ABTS and DPPH assays presented IC_50_ values of 115.2 mg/L and 544.0 mg/L, respectively. Furthermore, the extract displayed notable α-amylase inhibition and no cytotoxicity towards human fibroblasts. The primary phenolic compounds identified were catechin, epicatechin, and caffeic acid. Subsequently, MOLP and its extract were incorporated into bread and biscuits, replacing 5% of wheat flour, resulting in fortified functional foods. The fortified products exhibited improved TPC and antioxidant activity compared to the non-fortified foods. Furthermore, they displayed the ability to inhibit microbial growth, leading to an extended shelf life. Sensory analysis indicated that the products incorporated with the extract were preferred over those with MOLP. These results have demonstrated the viability of using MOLP and its phenolic-rich extract as an environmentally sustainable strategy for enhancing the quality of cereal-based products.

## 1. Introduction

In low- to middle-income countries, predominantly Africa and Southern Asia, monotonous diets consisting mainly of starchy foods are consumed since these are more easily accessible and affordable compared to other more nutritious sources that include fruits, vegetables, and animal-sourced foods. In these countries, food subsidies mostly involve starchy staples, including wheat flour, bread, and rice. On average, the cost of a healthy diet is 60% more than a diet that just meets the minimum dietary energy needs through a starchy staple; however, such diets are deficient in dietary diversity and can lead to micronutrient deficiencies—a significant public health concern [1]. Moreover, the limited shelf life of foods exacerbates food security and hunger issues in developing countries, where inadequate transportation and storage infrastructure result in significant food loss. This leads to food waste, economic losses for farmers, and limited access to nutritious food, negatively impacting communities [2].

With a growing urge to address these concerns and an increasing interest in leading a healthy lifestyle, functional foods have gained popularity and are now vastly studied and included in human diets [3,4]. Consequently, food fortification techniques have become vital in the food industry. Many functional ingredients are derived from natural sources, such as plants, which contain a high number of nutrients, vitamins, and compounds with beneficial properties for human health known as bioactive compounds (BACs) [5].

*Moringa oleifera* is a plant nutraceutical, given that it serves both as a nutritional and medicinal plant and can be incorporated as a natural food ingredient to produce functional foods. It belongs to the Moringacea family, which comprises 13 other species and is extensively cultivated across the Middle East, Africa, and Asia. Although originally from India, it has been widely cultivated and naturalised in many countries around the world, including tropical and subtropical regions of Asia, Africa, and South America [3,6]. Remarkably, the geographical distribution of this plant aligns with regions affected by malnutrition [7].

*M. oleifera* is commonly known as the miracle plant or tree of life and is believed to hold beneficial properties in all its parts, including leaves, roots, pods, seeds, and flowers, due to containing a high number of essential nutrients such as proteins, minerals, vitamins (A, B1, B2, B3, C, and E), and fibre [8]. Furthermore, the plant is known to possess a significant quantity of BACs, specifically phenolic compounds (PCs), tannins, and carotenoids, giving this plant antioxidant, antimicrobial, anti-inflammatory, hypoglycaemic, hepatoprotective, and anticancer properties [9,10]. The most common PCs present in *M. oleifera* include phenolic acids (e.g., caffeic acid, chlorogenic acid, gallic acid) and flavonoids (e.g., catequin, epicatequin, quercetin, kaempferol, rutin) [11].

PCs are secondary metabolites of plants that function as bio-antioxidants able to reduce reactive oxygen species (ROS) by donating hydrogen atoms of phenolic hydroxyls and by transferring electrons from such phenolic hydroxyls [12]. When incorporated into functional foods, besides helping prevent numerous diseases such as heart-related disorders and certain cancers, they also aid in prolonging their shelf life [13]. They achieve it by inhibiting oxidation, neutralising free radicals, delaying oxidative reactions, chelating metal ions, and regenerating other antioxidants, thus preserving the freshness, flavour, texture, and nutritional value of foods for a longer period of time [14]. This helps reduce food waste and enhances food security. PCs can be extracted from *M. oleifera* leaf powder (MOLP) using polar solvents such as ethanol since these compounds are relatively polar; however, the phenolic content in MOLP extracts varies depending on several factors, such as the plant’s origin and extraction conditions [15].

Furthermore, *M. oleifera* leaf and seed extracts have been shown to possess antidiabetic properties due to their inhibitory activity towards α-amylase and α-glucosidase enzymes responsible for breaking down carbohydrates into simple sugars [16]. Inhibiting these enzymes is a common approach to regulating blood glucose levels by slowing down the digestion of carbohydrates. Elevated blood glucose levels can trigger glycation, a process that causes secondary complications of diabetes, including impaired vision, heart disease, kidney failure, and more. Compounds that inhibit glycation have the potential to alleviate these complications by reducing the formation of advanced glycation end products and mitigating the associated problems related to oxidative stress, inflammation, and compromised immune function. Currently, diabetes management primarily involves oral hypoglycaemic medications, which can produce side effects such as lactic acidosis, gastrointestinal issues, and anaemia, therefore, alternatives (especially from food and vegetable sources) are necessary [17]. 

The interesting composition of *M. oleifera*, allied with the fact that it is a fast-growing tree that can thrive in poor soils and dry lands requiring little agricultural care, presents a sustainable approach to tackle malnutrition and food insecurity, as in providing access to safe, nutritious foods to meet their dietary needs for an active and healthy life by food fortification technique. 

To preserve consumer safety, it is crucial to ensure that the consumption of fortified products does not lead to any negative effects. Several studies were conducted in vitro and in vivo to explore the safety of MOLP and extracts, with most of them showing a high degree of safety. In vivo studies on rats showed that *M. oleifera* aqueous extracts were safe, presenting acute toxicity only at supra-supplementation levels (3 g/kg), being considered safe at levels ≤1 g/kg b.wt. (body weight), a dose that is still higher than the ones commonly consumed [18]. The same study indicated that concentrations of aqueous leaf extracts exceeding 20 mg/mL were found to be toxic to human peripheral blood mononuclear cells. It is worth noting, though, that achieving such concentrations through oral ingestion is not feasible. In another study, it was determined that ethanolic leaf extracts exhibited no adverse effects on fibroblasts within the concentration spectrum of 0.02 g/mL to 100 g/mL [19]. Lastly, no negative effects were observed in any human trials involving MOLP to date. Therefore, *M. oleifera* extracts seem to be safe for human consumption in the amounts commonly used [20]. Despite the existing findings, there is still a need for more toxicity studies to guarantee that supplements made from *M. oleifera* leaves are harmless for human health. There is a growing interest in directly including powders extracted from *M. oleifera* leaves or seeds as a natural food fortification in a range of cereal-based products, including bread, biscuits, cake, pasta, and various snacks. Furthermore, there is a rising awareness of the topic of food preservation, which warrants the usefulness of antioxidants to improve the quality of foods. The commercial availability of synthetic antioxidant compounds, such as butylated hydroxytoluene (BHT) and butylated hydroxyanisole (BHA), is limited due to their potential toxicity towards vital organs in the body, which pushes interest in the ongoing study and consumer preference for natural antioxidant compounds [21]. These natural alternatives, such as PCs, are generally considered safer and have garnered significant attention for their ability to effectively deliver antioxidant and antimicrobial properties to preserve food products and prevent the occurrence of rancidity caused by oxidation [21]. 

Most of the research on the fortification of bakery foods has been conducted using MOLP. Table 1 summarises the methods and findings of some studies conducted on the incorporation of MOLP in cereal-based products.

Numerous studies have confirmed the effectiveness of *M. oleifera* in enhancing the nutritional content of various cereal-based food products. Nevertheless, as far as the authors are aware, there is a notable absence of studies that have explored the utilisation of *M. oleifera* extracts, which contain a significantly higher concentration of BACs compared to the powdered form. Furthermore, the existing studies tend to focus more on the nutritional value of the plant and less on their antioxidant effectiveness, which aids in decreasing the rancidity process of foods and increasing their shelf life, a particularly important aspect in the case of bread since it presents a short shelf life of around 3–7 days. 

Therefore, the aim of this study was to produce bread and biscuits fortified with MOLP and its respective extract and assess their effect on the physicochemical properties of these foods, such as antioxidant capacity and microbial stability. A sensory assessment of the fortified bread and biscuits was conducted to evaluate the consumers’ demands and preferences for these novel products. 

## 2. Materials and Methods

### 2.1. Samples and Reagents

The *Moringa oleifera* leaf powder (MOLP) used during this work was provided by Agostinho Neto University (UAN). The leaves were collected in Luanda, Angola (8°57′24.9″ S, 13°13′02.9″ E), washed with running tap water, dried until obtaining a constant weight, and grounded to obtain a homogeneous powder (particle size < 250 μm). 

To produce the bread and biscuits, wheat flour (type 55), yeast, salt, eggs (about 60 g each), butter, and sugar were purchased from a supermarket in Porto, Portugal. 

Acetonitrile (Ref. 45983, C_2_H_3_N, CAS 75-05-8), ethanol (Ref. 83813.360, C_2_H_6_O, CAS 64-17-5), and methanol (Ref. 20834.291, C_3_H_2_O, CAS 67-56-1) were obtained from VWR (Radnor, PA, USA). Merck Millipore Mill-Q water purification equipment from Billerica (MA, USA), with 18.2 Ω of electric resistance, was used to obtain ultrapure water.

The sodium carbonate (Ref. 1.06392, CNa_2_O_3_, CAS 497-19-8) acquired from Merck (Darmstadt, Germany) and the Folin–Ciocalteu reagent (Ref. 47641) obtained from Sigma-Aldrich (St. Louis, MO, USA) were utilised for analysing the total phenolic content.

To assess the antioxidant capacity, Sigma-Aldrich reagents were used: 2,2-diphenyl-1-picrylhydrazyl (DPPH) (Ref. D9132, C_18_H_12_N_5_O_6_, CAS 1898-66-4) and 2,2′-azino-bis(3-ethylbenzothiazoline-6-sulfonic acid) (ABTS) (Ref. A1888, C_18_H_24_N_6_O_6_S_4_, CAS 30931-67-0).

For antibacterial evaluation, agar (Ref. J637, CAS 9002-18-0) and Plate Count Agar (Ref. 84608.0500) were obtained from VWR, m-Lauryl Sulfate Broth (Ref. 0734) was purchased from Sigma-Aldrich, and Rose-Bengal Chloramphenicol Agar (Ref. 1.00467.0500) was acquired from Merck. Sorbic acid (Ref. S1626, C_6_H_8_O_2_, CAS 110-44-1), which served as a positive control, was also acquired from Sigma-Aldrich. Dimethyl sulfoxide (Ref. 41640, C_2_H_6_OS, CAS 67-68-5) was obtained from Honeywell (Charlotte, NC, USA).

The α-amylase inhibition capacity was evaluated using α-amylase from porcine pancreas (Ref. A3176) and starch from corn (Ref. S4180, C_6_H_10_O_5_, CAS 9005-25-8) acquired from Sigma-Aldrich. 

For the cytotoxicity evaluation, Foetal Bovine Serum (FBS), Dulbecco’s Modified Eagle’s Medium (DMEM), trypsin, and Penicillin-Streptomycin (PenStrep) were purchased from Gibco (Cambridge, MA, USA).

The analytical standards used for phenolic compound quantification were acquired from Sigma-Aldrich: chlorogenic acid (Ref. 1115545, C1_6_H_18_O_9_, CAS 327-97-9), gallic acid (Ref. 147915, C_7_H_6_O_5_, CAS 149-91-7), catechin (Ref. 43412, C_15_H_14_O_6_, CAS 154-23-4), kaempferol (Ref. 60010, C1_5_H_10_O_6_, CAS 520-18-3), caffeic acid (Ref. C0625, C_9_H_8_O_4_, CAS 331-39-5), epicatechin (Ref. E1753, C_15_H_14_O_6_, CAS 490-46-0), and quercetin (Ref. Q4951, C_15_H_10_O_7_, CAS 117-39-5).

### 2.2. Extraction of Bioactive Compounds from Moringa oleifera

Phenolic compounds from *Moringa oleifera* leaf powder (MOLP) were obtained through a solid–liquid extraction method utilising a Soxhlet apparatus, as reported in the literature [27]. The extraction extended over a 2 h period, employing ethanol as the solvent, with a sample-to-solvent ratio of 1:40 (*m*/*v*). Following this, solvent evaporation took place using a rotary evaporator (Rotavapor R-200, BUCHI Laboratories, Flawil, Switzerland), followed by a continuous nitrogen stream at 2 mbar.

### 2.3. Characterisation of Moringa oleifera Extract

#### 2.3.1. Total Phenolic Content and Antioxidant Capacity

The assessment of the total phenolic content (TPC) in the extract was conducted using the Folin–Ciocalteu method, following the procedure outlined in the literature [28]. A solution of the extract at a concentration of 1000 mg/L was incubated with the Folin–Ciocalteu reagent and a sodium carbonate solution (333.3 g/L) for a duration of 2 h at room temperature. Following that, the absorbance was recorded at 750 nm, and the results were expressed in milligrams of gallic acid equivalents (GAE) per gram of the extract. The antioxidant capacity of the MOLP extract was assessed through the DPPH and ABTS assays, according to literature procedures with minor modifications [29]. In the ABTS test, the extract solutions (ranging from 100 mg/L to 2500 mg/L) were incubated with the ABTS solution for 15 min, and the absorbance was measured at 734 nm. For the DPPH assay, the extract solutions (ranging from 1500 mg/L to 8000 mg/L) were incubated for 40 min with the DPPH solution, after which the absorbance was measured at 515 nm. In both assays, a curve of the percentage inhibition of free radicals vs. the extract concentration was constructed, and the IC_50_ values were determined. The results were also expressed in milligrams of Trolox equivalents (TE) per gram of the extract. All measurements were performed in triplicate.

#### 2.3.2. Antibacterial Capacity

The antibacterial activity of the extract was tested against *Escherichia coli* and *Staphylococcus aureus* bacteria using the disk diffusion test in the biological culture medium Plate Count Agar (PCA). Two different concentrations of the extract (500 mg/mL and 1000 mg/mL) were prepared in 2% aqueous DMSO. A common food preservative, sorbic acid, was used as the positive control (500 mg/mL and 1000 mg/mL) while ultrapure water was used as the negative control. The test was performed according to the protocol described by Ferreira and Santos [28]. Bacterial suspensions of each microorganism, with an optical density of 0.1 at 610 nm, were plated on PCA medium. Afterwards, 7 μL of the extract or ascorbic acid solution (at both concentrations) were added to a sterile 5 mm diameter disc in triplicate. The same was performed for the negative control. After incubating the plates for 24 h at 37 °C, the inhibition levels were measured in triplicate.

#### 2.3.3. α-Amylase Inhibition 

The extract was tested for its inhibitory potential using a modified α-amylase assay, following the procedures described in reference [30]. In summary, 250 µL of the extract solution (1 mg/mL in ethanol) were mixed with 250 µL of α-amylase solution (100 mg/L in ultrapure water) and incubated at 37 °C for 10 min. Subsequently, 250 µL of starch solution (1% *w*/*v*) were added, followed by another 10 min incubation at 37 °C. Then, 500 µL of 3,5-dinitrosalicylic acid (DNS) reagent were added, and the samples were heated in boiling water for 10 min. After cooling, 5 mL of ultrapure water was added, and the absorbance was measured at 540 nm. The measurements were performed in triplicate.

#### 2.3.4. Phenolic Compounds Quantification

This study used high-performance liquid chromatography with a diode array detector (HPLC-DAD) to identify and quantify phenolic compounds in the *M. oleifera* extract. An Elite LaChrom HPLC system (Hitachi, Tokyo, Japan) with a Puroshper^®^ STAR RP-18 endcapped LiChroCART^®^ column (Merck, Germany) was utilised. Sample solutions were prepared using an acetonitrile:water:ethanol solvent mixture (2:1:1 *v*/*v*/*v*). The mobile phase consisted of two eluents: Milli-Q water with 0.5% orthophosphoric acid (A) and methanol:acetonitrile (80:20 *v*/*v*) (B). The gradient ranged from 10% to 70% B over 60 min [31]. Compounds were detected at specific wavelengths: catechin and epicatechin were detected at 222 nm, gallic acid at 275 nm, and caffeic acid, along with chlorogenic acid, at 322 nm. Additionally, kaempferol and quercetin were identified at 365 nm. Quantification relied on calibration curves and the external standard method, with spike samples aiding peak identification and quantification. All measurements were performed in triplicate.

#### 2.3.5. Cytotoxicity

The cytotoxicity of the obtained extract was evaluated against HFF-1 human fibroblasts using the resazurin assay. The cells were cultured in DMEM supplemented with 10% FBS (*v*/*v*) and 1% PenStrep (*v*/*v*) at 37 °C in a 5% CO_2_ atmosphere. Briefly, 200 μL of HFF-1 cells were seeded in a 96-well plate at a density of 13,000 cells/well and incubated for 24 h at 37 °C, 5% CO_2_. Afterwards, 40 μL of the medium were replaced by the extract solution (5000 mg/L in ultrapure water), and the plate was incubated overnight at 37 °C with 5% CO_2_. Non-treated cells (cells without the addition of any compound) were used as a negative control, while hydrogen peroxide was used as a positive control since it is an oxidising agent capable of causing cell death. After incubation, the medium was replaced by 200 μL of resazurin (20% *v*/*v*), and the plate was incubated for 4 h, protected from the light, at 37 °C with 5% CO_2_. Finally, the fluorescence was assessed using the microplate reader Synergy HT (BioTek Instruments Inc., Winooski, VT, USA) at an excitation wavelength of 530 nm and an emission wavelength of 590 nm. The cell viability percentage was then calculated in reference to the untreated cells. Four independent measurements were performed in triplicate.

### 2.4. Incorporation of Moringa oleifera Leaf Extract in Bread and Biscuits

#### 2.4.1. Production of Breads and Biscuits

To evaluate the effect of the incorporation of MOLP and extract in bread (Br) and biscuits (Bi), three formulations of each food were prepared: a negative control (NC) with no additional ingredients; a formulation with MOLP at a wheat flour substitution level of 5% *w*/*w* (MP); and a formulation with the extract at a wheat flour substitution level of 5% *w*/*w* given the extraction efficiency (ME). The ingredients of each formulation are detailed in Table 2 and Table 3. 

Breads were produced using the Home Bread machine from Tefal (Sarcelles, France) using program 3 (French bread). After production, each bread loaf was sliced and sealed in separate plastic bags. Biscuits were produced using the Thermomix machine (Vorwerk, Wuppertal, Germany). The ingredients were mixed following the instructions, and afterwards the biscuits were left to bake for 8 min at 200 °C in a convection oven from Gaggenau (Gaggenau, Germany). The biscuits were also stored in separate plastic bags. The bags were stored at room temperature and kept in the dark to avoid the breakdown of the bioactive compounds present in *M. oleifera* since they are photosensitive. 

#### 2.4.2. Antioxidant Capacity

Subsequently, the antioxidant properties of the bread and biscuits were examined by extracting the phenolic compounds present in each formulation using ethanol. Briefly, 2 g of sample were subjected to an extraction process with the addition of 4 mL of ethanol, followed by three repetitions of vortexing (1 min) and ultrasonic bathing (5 min). The resulting solutions underwent centrifugation for 20 min at 3000 rpm using a Rotofix 32 A centrifuge (Hettich, Tuttlingen, Germany). The collected supernatant was then subjected to an additional round of 4 mL of ethanol addition and a repetition of the entire process. Finally, the resulting supernatant was utilised for the analysis of the antioxidant capacity (ABTS and DPPH) of the various formulations, following the methodologies delineated in Section 2.3.1. The bread was analysed the day after production (t0), four days after production (t1), and one week after production (t2). The biscuits were analysed the day after production (t0), two weeks after production (t1), four weeks after production (t2), and seven weeks after production (t3). All measures were performed in triplicate.

#### 2.4.3. Microbial Analysis

For this assay, two different mediums were used: Lauryl Sulphate Agar (LSA), selective to coliform microorganisms, and Rose Bengal Chloramphenicol Agar (RBC), selective to yeast and moulds. The analysis was performed according to the literature [27]. The bread crumb/biscuits were mixed in 9 mL of saline solution (0.9% NaCl), followed by 1 min of vortexing. Subsequently, a volume of 100 μL from each solution was inoculated onto the respective medium. The LSA plates underwent a 37 °C incubation for 24 h, while the RBC plates were incubated for 7 days at 25 °C. Subsequently, the plates were examined for the presence of microorganisms. The analysis was performed in duplicate.

#### 2.4.4. Sensory Assessment

For the sensory evaluation, the freshly made breads were sliced into individual portions, one for each participant, who were habitual bread and biscuit consumers and ranged from 20 to 63 years of age. The panel consisted of 20 untrained panellists: 12 females and 8 males. Using a 5-point hedonic scale (1: dislike extremely, 3: neither like nor dislike, and 5: like extremely), the bread’s visual appeal, colour, flavour, aroma, texture, and overall acceptability were assessed. The same reasoning was applied to the biscuits for their sensory assessment.

### 2.5. Statistical Analysis

The statistical analysis was conducted using GraphPad Prism 8.0.2, employing an analysis of variance (ANOVA) and Tukey’s multiple comparisons test. Statistically significant differences were considered for values with *p* < 0.05 at a 95% confidence interval. The results were expressed as mean ± standard deviation.

## 3. Results and Discussion

### 3.1. Characterisation of Moringa oleifera Extract

In this study, the phenolic compounds (PCs) from M. oleifera leaf powder (MOLP) were extracted using a solid–liquid extraction technique with a Soxhlet apparatus. Soxhlet extraction offers several appealing advantages over other extraction methods. Its simplicity requires only low-cost equipment, facilitating the extraction of large quantities of material without the need for subsequent filtration. Despite its downsides, such as the prolonged extraction time and solvent usage, this extraction method continues to be extensively employed in many laboratories [32]. Recent advancements in extraction techniques have shifted the focus towards more environmentally friendly and sustainable solutions. Techniques such as supercritical extraction and ultrasound-assisted extraction (UAE) have gained popularity [33], together with microwave-assisted extraction (MAE) and pressurised liquid extraction (PLE). These techniques often yield very rich phenolic extracts due to the possibility of using moderate temperatures and short extraction times, reducing the probability of degradation of PCs and lowering the environmental footprint.

Table 4 comprises the extraction yields obtained in different studies regarding the extraction of PCs from MOLP using various extraction techniques. Various factors, including the extraction method, solvent, time, temperature, and sample mass-to-solvent volume ratio, can influence the extraction yield. In the current study, an extraction yield of 41.4 ± 11.8% was obtained, a value superior to most of the previously reported studies, showing the efficiency of the Soxhlet method in extracting phenolic compounds.

The phenolic-rich extract obtained in the present work was characterised regarding its total phenolic content (TPC) and antioxidant, antimicrobial, and anti-diabetic properties. The results are presented in Table 5.

The TPC and antioxidant capacity are highly influenced by many parameters, such as the plant source, geographical location, climate, and growing conditions. The timing of harvest, extraction method, storage conditions, and chemical composition of PCs also play a role [35]. From Table 5, it is possible to observe that the extract obtained in the current work presented an average TPC value of 138.2 mg_GAE_/g_extract_. Concerning the antioxidant capacity, the results were expressed both in IC_50_ values, which represent the extract concentration required to inhibit 50% of the free radicals (DPPH or ABTS), and in comparison with Trolox (TEAC), a standard antioxidant compound. The smaller the value of IC_50_, the less amount of extract is needed to inhibit the radicals to the same extent, therefore the stronger the antioxidant capacity. In the case of the TEAC, the higher the value, the stronger the antioxidant capacity. The *M. oleifera* extract was able to inhibit both radicals; however, the extract presented a higher antioxidant capacity towards ABTS since its IC_50_ value is much lower compared to DPPH (115.2 ± 4.9 mg/L vs. 544.0 ± 7.9 mg/L) and its TEAC is higher. This higher capacity to inhibit ABTS compared to DPPH was also described by other researchers [28,36]. In another study, the exact same extraction method and conditions were used to extract PCs from MOLP. The results of the IC_50_ of DPPH and ABTS assays were quite similar to those of this study: 636.0 ± 9.2 mg/L and 205.2 ± 4.6 mg/L, respectively; however, the TPC (79 mg_GAE_/g_extract_) was almost half of the one obtained in the present study, suggesting that a higher TPC does not necessarily mean a higher antioxidant capacity [27]. A similar pattern was noted in another study employing ultrasound-assisted solid–liquid extraction, where MOLP extract exhibited a reduced TPC at 54.5 ± 16.8 mg_GAE_/g and higher antioxidant capacity, reflected in lower IC_50_ values (133.4 ± 12.3 mg/L for DPPH and 60.0 ± 9.9 mg/L for ABTS) [28]. In a different study, similar extraction conditions as the ones selected for the present work were used, with the exception of the extraction solvent (70% acetone). The extract obtained had a TPC of 26.07 ± 0.80 mg_GAE_/g. These results suggest that using ethanol as compared to acetone might be more fitting to extract phenolic compounds from *M. oleifera* leaves [37]. Other studies have reported that using a mixture of ethanol and water as the extraction solvent usually tends to yield a higher number of TPC in the extracts when compared to using ethanol alone [38,39]. These findings suggest that the choice of extraction solvents may impact both the phenolic composition and antioxidant properties of *M. oleifera* extracts. 

Regarding the antibacterial activity, there were no inhibition halos identified in the *M. oleifera* leaf extract sample. These results should not be interpreted as the extract’s inability to hinder the growth of *E. coli* and *S. aureus*, as the halo may still be present below the disk. While other research highlights the antibacterial efficacy of *M. oleifera* leaf extract against these microorganisms, it is crucial to acknowledge that the concentrations investigated in those studies significantly exceeded the ones examined in our research. This divergence in concentration levels might account for the discrepancies observed in the obtained results [40]. Furthermore, the source of the *M. oleifera* leaves, the extraction method, the inhibition medium, and the reagents used also have an impact.

In this study, the mean percentage of α-amylase inhibition was 94.1 ± 0.4%. Other studies have also proven the efficacy of *M. oleifera* leaf extract in inhibiting this enzyme. A particular study found that the leaves have the highest percentage of α-amylase inhibition when methanolic solvents are used (65.6 ± 4.93%), followed by other extraction solvents (hexane: 52.3 ± 2.5%, distilled water: 43.3 ± 2.3%, ethanol: 33 ± 2.6%) [17]. In another report, the authors stated that the methanolic leaf extract (5 mg/mL) presented a percentage enzyme inhibition of 50.6 ± 5.9%, followed by hexane (47.3 ± 2.8%) at the same concentration [41]. The α-amylase inhibitory effects may be attributed to the extracts’ richness in a diverse number of phytochemical and antinutritional factors, inducing an optimistic path for creating innovative cereal-based products that can help control chronic illnesses such as type II diabetes [13,41].

To have a better understanding of the biological characteristics demonstrated by the *M. oleifera* extract, an HPLC-DAD examination was conducted to identify and quantify the principal phenolic compounds in the obtained extract. As indicated in Table 6, it is evident that catechin emerged as the predominant phenolic compound in the extract, trailed by other flavonoids like epicatechin and caffeic acid. Residual amounts of chlorogenic acid and quercetin were also found. Gallic acid and kaempferol were not detected. 

Previous studies have already reported the presence of these compounds in *M. oleifera* leaf extract [42]. The literature reports show some variability in the values obtained due to factors such as the origin and cultivation conditions of *M. oleifera* trees, as well as the extraction method used to isolate phenolic compounds, which can influence their concentration. In a recent study, the same PCs as this study were identified and quantified. Catequin, epicatechin, caffeic acid, chlorogenic acid, and quercetin were found at concentrations of 19.83, 0.67, 0.16, 1.04, and 0.06 mg_compound_/g_extract_, respectively [43]. Other studies have also reported higher concentrations of PCs. For example, in a separate study that examined seven phenolic compounds, the concentrations extended from 19.65 mg/g for kaempferol to 65.83 mg/g for chlorogenic acid [44]. Likewise, in another study, the concentrations of chlorogenic acid, kaempferol, epicatechin, quercetin, and gallic acid were also higher than those obtained in this research [45]. Moreover, it is crucial to highlight that while some authors acknowledged gallic acid as a primary compound in *M. oleifera* leaves [46], this compound was not found in the analysed extract. This aligns with the observations of other researchers who similarly did not detect gallic acid [47]. One possible explanation for this could be that gallic acid is a very polar phenolic compound and, thus, potentially challenging to extract using ethanol or hydroalcoholic solutions with low water content. Nevertheless, the results obtained in the present work suggest that the presence of the phenolic compounds identified in the *M. oleifera* leaf extract can explain its antioxidant and antidiabetic properties.

Finally, cytotoxicity studies were performed to confirm the *M. oleifera* extract’s safety for human consumption. An extract’s concentration of 1000 mg/L was selected, taking into consideration that it is a concentration that presents great biological properties (as seen in Table 5). The results are presented in Figure 1. 

When in contact with cells, a substance can be considered non-cytotoxic when the cells present a viability superior to 70% [48]. As shown in Figure 1, the *M. oleifera* extract presented similar metabolic activity (not statistically different) compared to the non-treated cells and was well above the limit of 70%. This result demonstrated that, at the concentration tested, the extract is safe to be used in human applications, such as its incorporation in cereal-based products. Other studies have also accessed the safety of *M. oleifera* leaf extracts in fibroblasts, proving that the ethanolic extract is safe at concentrations from 0.02 g/mL to 100 g/mL [19], which are much higher than the one tested in this study (0.001 g/mL). According to another report, fibroblasts remained unaffected by aqueous and methanolic extracts when concentrations below 0.5 mg/mL were examined [49]. Regarding the present work, further analysis is required, using higher concentrations, to determine the limit of incorporation that is safe and does not pose any risk to human health.

### 3.2. Characterisation of Fortified Breads and Biscuits

#### 3.2.1. Physical Characterisation

To analyse the feasibility of using MOLP and its extract to create value-added cereal-based products, three different formulations of bread and biscuits were prepared. A negative control bread/biscuit without *M. oleifera* incorporation (NC-Br/NC-Bi), a bread/biscuit with a 5% substitution level of wheat flour for MOLP (MP-Br/MP-Bi), and a third bread/biscuit with substituted wheat flour for *M. oleifera* extract corresponding to a 5% MOLP substitution taking into consideration the extraction yield (ME-Br/ME-Bi). The visual appearance of the different formulations produced is shown in Figure 2 and Figure 3.

The fortified breads and biscuits presented a greenish hue due to the dark green colour of the MOLP and its respective extract, with the bread fortified with MOLP (MP-Br) appearing to be darker (Figure 2B). It was also shown to be denser, shorter, and less voluminous compared to the regular bread (NC-Br) (Figure 2A) and the bread fortified with MOLP extract (ME-Br) (Figure 2C). These findings were also noted by other authors, who attributed these observations to the powder’s binding properties and moisture retention. Compounds in the powder can interact with gluten, hindering dough expansion during fermentation and leading to a more compact crumb structure. Additionally, the powder’s moisture absorption competes with yeast activity, affecting the rise. The reduction in both height and volume might also result from the antimicrobial effects of *M. oleifera* leaves on the yeast’s leavening activities while the dough ferments [50]. 

Regarding the biscuits, there was no clear visible difference in colour between MP-Bi and ME-Bi (Figure 3), neither the volume nor density of the cookies after baking. However, other researchers have noted that the progressive inclusion of MOLP from 0 to 10% resulted in a reduction in the spread ratio in biscuits due to reduced diameter and increased thickness, linked to gluten dilution and decreased water available for gluten hydration. The authors argued that adding composite flour increased dough viscosity and formed aggregates by competing for limited free water in the cookie dough [51].

#### 3.2.2. Total Phenolic Content and Antioxidant Capacity

To understand the impact of the incorporation of MOLP and its phenolic-rich extract on the breads’ and biscuits’ properties, the TPC and antioxidant capacities of the foods were analysed over time. The results obtained for the TPC of the formulations at different timepoints are presented in Figure 4.

It is clear that the fortification of both the bread and biscuits with *M. oleifera* led to a substantial rise in the phenolic content of the formulations throughout the entire duration of this study. This outcome was anticipated, given the rich abundance of phenolic compounds (PCs) present in *M. oleifera* leaves. Other authors had similar findings [23,51]. Moreover, the TPC of the foods fortified with MOLP (MP-Br and MP-Bi) presented a higher value at all the timepoints of this study in comparison to the extract-incorporated formulations (ME-Br and ME-Bi). This finding can be due to the fact that the MOLP extraction might have led to the degradation or loss of some of the PCs, thereby reducing their concentration in the extract compared to the original powder. This could have led to a dilution effect where the PCs in the extract were spread out over a larger volume of bread, potentially resulting in lower measured TPC compared to the powder. 

Phenolic compounds are susceptible to degradation from factors like light, heat, and oxygen, which can result in a decrease in their content over time [52]. It is noticeable from Figure 4 that both the fortified breads and biscuits TPC decreased over time. Regarding the bread (Figure 4A), when comparing MP-Br and ME-Br, the first one had a significantly higher phenolic content right after production. However, with time, this initial difference in phenolic content between MP-Br and ME-Br seemed to diminish, and after one week, both formulations exhibited nearly identical (non-significantly different) TPC values. This suggests that ME-Br has the interesting property of retaining its PCs more effectively over time. The same happened with biscuits (Figure 4B).

Regarding the antioxidant properties of the breads and biscuits, the results for both the DPPH and ABTS assays for both foods at the different timepoints are displayed in Figure 5. 

From Figure 5, it is clear that the fortification of bread and biscuits increased their antioxidant capacity throughout the entire period of this study due to showing a substantially higher percentage inhibition for both radicals, with the ABTS assay having higher values, as expected (Figure 5C,D). Similar to the TPC assay, the MP-Br and MP-Bi formulations presented a higher antioxidant capacity in comparison to the extract-incorporated foods. While there is a lack of existing literature on the inclusion of *M. oleifera* extracts into bread, previous studies, as mentioned before, have explored the enhancement of bread through the addition of MOLP. These studies consistently noted a boost in the antioxidant capacity of the bread with increasing levels of fortification [23,24]. Nevertheless, an analysis of these properties over time was not conducted.

Analysing Figure 5A,C, it is noticeable that for the three breads, the antioxidant capacity had the tendency to decrease over time; however, in the DPPH assay, ME-Br showed once again that it can retain the PCs more efficiently than MP-Br. Regarding the biscuits (Figure 5B,D), the values of their antioxidant capacity were lower in comparison to the breads due to their containing a larger number of different ingredients, which in turn creates a dilution effect on the PCs present in the samples. Other authors have also proven the increasing level of antioxidant capacity of biscuits with increasing levels of MOLP substitution. In the study conducted by Fapetu et al., the biscuit with a 5% wheat flour substitution for MOLP presented a DPPH inhibition of 40.42%, which is close to the value presented in this study of 35.37% [25]. Although it is not so common to incorporate MOLP extract in biscuits, some authors have tested the inclusion of 1% MOLP extract and also stated that it caused an increase in the antioxidant activity of the biscuits [53].

#### 3.2.3. Microbial Contamination

The presence of microorganisms, more specifically coliforms (using LSA medium), yeast, and moulds (using RBC medium), was investigated in each food formulation at different time periods. The results are presented in Table 7.

As shown in Table 7, the three bread formulations had no significant contamination the day after production (t0) in both LSA and RBC mediums. Considering the typical 4-day shelf life of bread at room temperature (t1), it was expected to find microbial contamination by the fourth day. As observed, the negative control bread (NC-Br) displayed a substantial number of colony-forming units (CFU) in both mediums, aligning with expectations. However, in the LSA medium, the bread with MOLP (MP-Br) presented a high degree of microbial contamination, which was somewhat unexpected since MOLP-enriched bread had the highest TPC and should theoretically delay oxidative spoilage, which in turn hinders microbial growth. In contrast, ME-Br exhibited impressive results, with only 3.05 × 10^5^ CFU/mL, which is lower than the unsatisfactory limit established by the Centre of Food Safety, Department of Food and Environmental Hygiene, categorising an upper limit of microbial concentration equal to ≥10^6^ CFU/mL for bakery and confectionery products [54]. This outcome underscores the extract’s effectiveness as a potent food preservative, validating its role in extending shelf life and maintaining product quality. A possible explanation for these findings is the fact that although MP-Br contained a higher phenolic content value than ME-Br, the MOLP extract might contain other compounds that have been proven to have an antimicrobial effect (such as alkaloids and glycosides) in a higher concentration, which aid in delaying the microbial growth in the bread [55].

Analysing yeast and mould growth results (RBC medium), there was a substantial growth of colonies for the NC-Br four days after production (t1) (>30 × 10^5^ CFU/mL) and no significant growth of colonies for the fortified bread samples at any timepoint. These results show that both the MOLP and its extract were able to efficiently inhibit microbial contamination in the food samples owing to their remarkable anti-microbial effect and antioxidant capacity, therefore having the ability to extend their shelf life. Other studies have also proven the efficacy of the incorporation of MOLP in inhibiting the growth of microorganisms, yeasts, and moulds in bread [56].

Regarding the biscuits, since they typically have an extended shelf life, there was no significant contamination in any of the mediums or timepoints, which suggests that this experiment should be conducted in a prolonged time frame in order to capture the potential growth of microbials. Other authors, however, have noted an increase in shelf life with MOLP-incorporated biscuits and crackers [57,58].

#### 3.2.4. Sensory Assessment of the Breads and Biscuits

A sensory assessment was conducted to measure consumers’ acceptability of bread and biscuits fortified with MOLP and its respective extracts. The obtained results are expressed in Figure 6. 

The sensory evaluation showed that the control bread and biscuits, as expected, had the highest scores in all categories. The decrease in scores of the fortified foods was likely due to the herbaceous taste of the leaves and noticeable colour changes in the foods, which differed from their familiar golden-brown appearance. Previous studies have noted that as MOLP levels increased in bread and biscuits (at 5% and 10% replacement levels), overall acceptability decreased, with brown bread being less affected than white bread [24,57,59,60]. 

The bread with *M. oleifera* extract (ME-Br) received a more positive response from consumers compared to the one incorporated with MOLP (MP-Br), as observed in Figure 6A. This preference could be attributed to the extract’s milder taste and lighter green colour. Regarding crumb texture, NC-Br and ME-Br received high scores, while MP-Br had decreased texture acceptability since MOLP increases bread hardness and gumminess due to its high fibre content [61]. Fibre absorbs water and swells during baking, resulting in a firmer and denser texture. Regarding the biscuits, the extract-incorporated biscuit (ME-Bi) was also better accepted than the powder-incorporated biscuit (MP-Bi), as seen in Figure 6B. Moreover, in general, the biscuits were better received than the bread in all categories due to the fact that the biscuits contain sweeter ingredients in their formulation that are able to mask the herbaceous taste and aroma of *M. oleifera* more effectively. 

In summary, the control bread and biscuits had the highest overall acceptability, the extract-enriched bread and biscuits scored slightly lower but still favourably, and the MOLP-incorporated foods had a moderate level of acceptability, noting that the biscuits had a higher score in all categories in comparison to the bread. Therefore, this study suggests that incorporating *M. oleifera* leaf extract in cereal-based foods is a more promising approach to offering the numerous benefits of this plant while still keeping in mind consumers’ taste preferences and satisfaction. 

## 4. Conclusions

This work is intended to evaluate the feasibility of utilising *Moringa oleifera* leaf powder (MOLP) and its phenolic-rich extract, renowned for their antioxidant capabilities, to enhance the quality of cereal-based products, with a specific focus on bread and biscuits. The results suggested that the extract derived from MOLP exhibited remarkable antioxidant and antidiabetic properties, highlighting its potential as a functional food ingredient for managing blood sugar levels. Regarding the cytotoxicity studies, these demonstrated that an extract’s concentration of 1000 mg/L is safe for human consumption. Furthermore, the fortified breads and biscuits, enriched with MOLP and its extract, consistently demonstrated an elevated total phenolic content and antioxidant capacity, outperforming the negative control over the study duration. Moreover, the application of both MOLP and its extract effectively limited microbial growth during the storage of food products, emphasising their dual role in enhancing food preservation. Moreover, there was an evident preference for extract-infused foods over powder-infused alternatives among consumers. The results with a 5% substitution level were quite promising, and as future work, it would be interesting to study higher levels to understand what the technological limit for the supplementation level is as well as the effect on the sensory properties. In conclusion, the potentiality of food fortification using MOLP and its phenolic-rich extract is a novel approach to the food industry to produce value-added products that offer richer nutrient composition, potential disease-preventing benefits, and an extended shelf life to tackle food insecurity and malnutrition. 

Despite being promising, the use of nutraceuticals such as *Moringa oleifera* in foods must be carefully analysed to define a safe dosage for consumers. Although a cytotoxicity analysis was performed in the present work, it is important to perform a more detailed analysis on the safety of the extract, for example, to evaluate potential allergens. Moreover, conducting a nutritional assessment of bread and biscuits, both prior to and following the inclusion of the powder and extract, would provide insights into their impact on nutritional content. Lastly, an evaluation of the effect of *M. oleifera* addition on the physical parameters of the products (such as colour and texture) would contribute to a thorough understanding of the sensory attributes of these products.

## Figures and Tables

**Figure 1 antioxidants-12-02069-f001:**
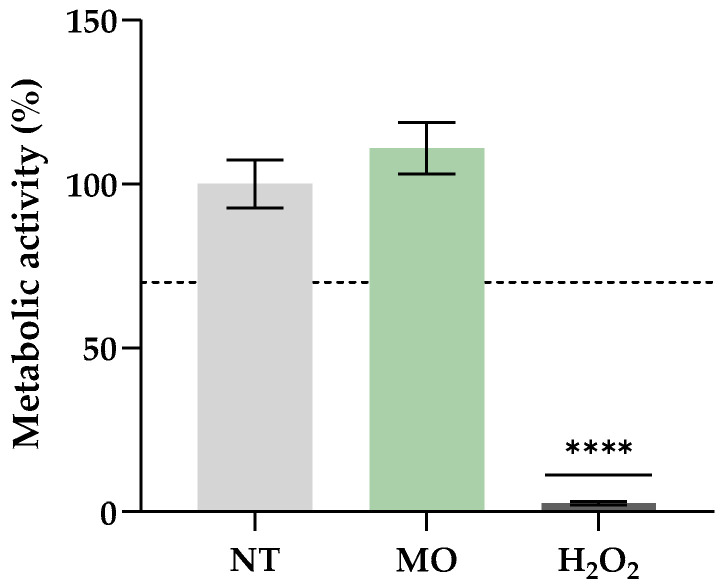
Cytotoxicity results of *Moringa oleifera* extract towards HFF-1 cells (human fibroblasts). The dashed line corresponds to 70% of cell viability. The results are expressed as means ± standard deviations of 4 independent measurements. NT—non-treated cells (negative control); MO—cells incubated with *M. oleifera* extract; H_2_O_2_—cells incubated with hydrogen peroxide (positive control). ****—statistically different values (*p* < 0.0001) in comparison to NT cells.

**Figure 2 antioxidants-12-02069-f002:**
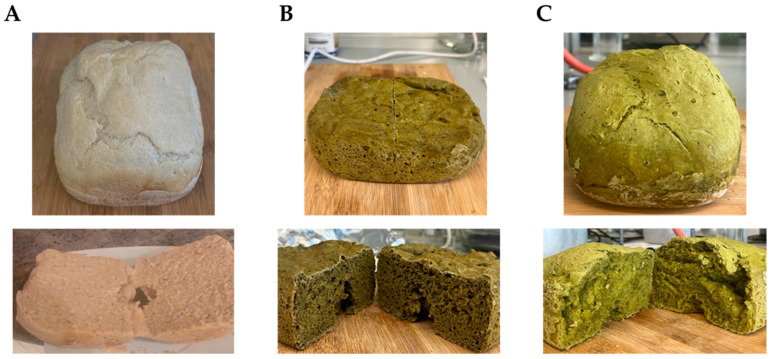
Visual appearance of bread’s crust (**top**) and crumb (**down**). (**A**) NC-Br—bread with no additives (negative control); (**B**) MP-Br—bread with 5% MOLP; and (**C**) ME-Br—bread with extract corresponding to 5% MOLP.

**Figure 3 antioxidants-12-02069-f003:**
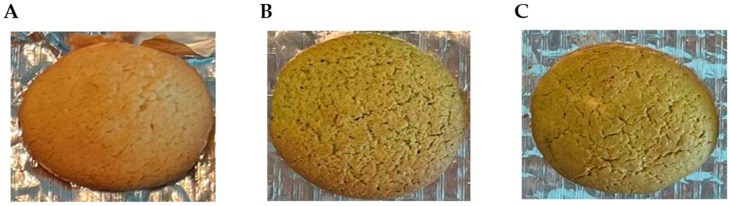
Visual appearance of biscuits’ formulations. (**A**) NC-Bi—biscuits with no additives (negative control); (**B**) MP-Bi—biscuits with 5% MOLP; and (**C**) ME-Bi—biscuits with extract corresponding to 5% MOLP.

**Figure 4 antioxidants-12-02069-f004:**
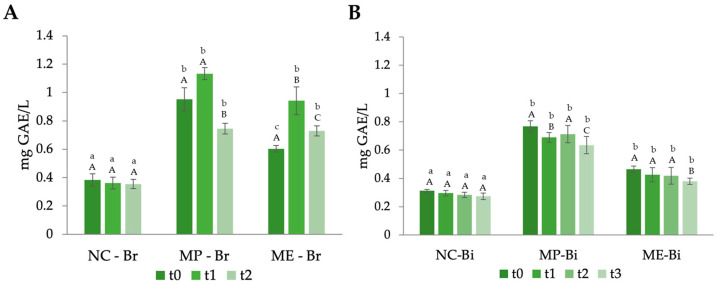
Total phenolic content variation in the breads (**A**) and biscuits (**B**) throughout the period of the study. NC-Br—bread with no additives (negative control); MP-Br—bread with 5% MOLP; ME-Br—bread with extract corresponding to 5% MOLP. NC-Bi—biscuits with no additives (negative control); MP-Bi—biscuits with 5% MOLP; ME-Bi—biscuits with extract corresponding to 5% MOLP. The analysis was performed at different timepoints: for bread, t0—day after production, t1—four days after production, and t2—one week after production; for biscuits, t0—day after production, t1—two weeks after production, t2—four weeks after production, and t3—seven weeks after production. The results are expressed as means ± standard deviations of 4 independent measurements. Different lowercase letters (a–c) represent statistically different values (*p* < 0.05) for the same timepoint. Different capital letters (A–C) represent statistically different values (*p* < 0.05) for the same formulation.

**Figure 5 antioxidants-12-02069-f005:**
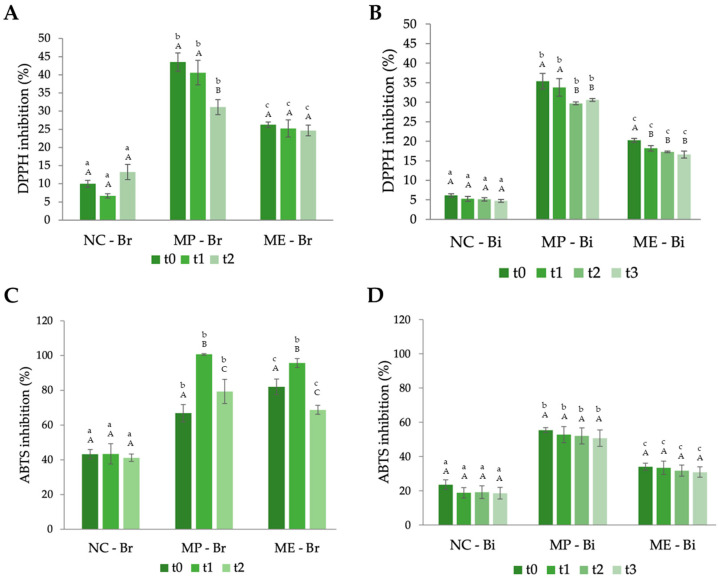
Antioxidant property variation in the breads and biscuits throughout the period of this study. (**A**) DPPH assay results for bread; (**B**) DPPH assay results for biscuits; (**C**) ABTS assay results for bread; and (**D**) ABTS assay results for biscuits. NC-Br—bread with no additives (negative control); MP-Br—bread with 5% MOLP; ME-Br—bread with extract corresponding to 5% MOLP. NC-Bi—biscuits with no additives (negative control); MP-Bi—biscuits with 5% MOLP; ME-Bi—biscuits with extract corresponding to 5% MOLP. The analysis was performed at different timepoints: for bread, t0—day after production, t1—four days after production, and t2—one week after production; for biscuits, t0—day after production, t1—two weeks after production, t2—four weeks after production, and t3—seven weeks after production. The results are expressed as means ± standard deviations of 4 independent measurements. Different lowercase letters (a–c) represent statistically different values (*p* < 0.05) for the same timepoint. Different capital letters (A–C) represent statistically different values (*p* < 0.05) for the same formulation.

**Figure 6 antioxidants-12-02069-f006:**
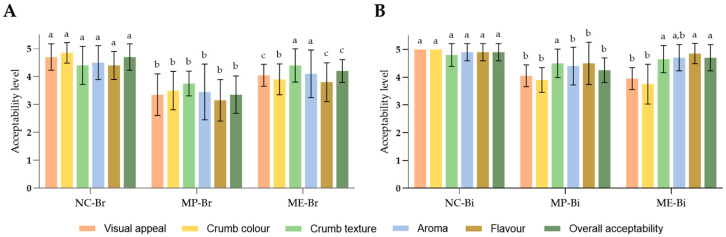
Results from the sensory assessment of bread (**A**) and biscuits (**B**). NC-Br—bread with no additives (negative control); MP-Br—bread with 5% MOLP; ME-Br—bread with extract corresponding to 5% MOLP. NC-Bi—biscuits with no additives (negative control); MP-Bi—biscuits with 5% MOLP; ME-Bi—biscuits with extract corresponding to 5% MOLP. The results are expressed as means ± standard deviations. Different lowercase letters (a–c) represent statistically different values (*p* < 0.05) for the same sensory parameter.

**Table 1 antioxidants-12-02069-t001:** Studies on the incorporation of *Moringa oleifera* in cereal-based food products.

Product	Study Methods	Findings	Ref.
Dried pasta	Formulation of pasta containing semolina and 10%, 20%, 30%, and 40% of MOLP. The moisture, protein, and ash content of the samples were investigated. A sensory evaluation and a texture profile analysis were also conducted.	The pasta fortified with MOLP exhibited higher protein and ash contents in comparison to the control pasta and a lower moisture content. Regarding textural parameters and consumer acceptability, the pasta with 20% MOLP inclusion was found to be the best.	[22]
Bread	Assessment of the impact of MOPL addition at levels ranging from 0% to 10% on the proximate, mineral, antioxidant, and sensory attributes of WWF-leavened bread.	The MOLP-supplemented bread showed noticeable improvements in proximate and mineral profiles. The TPC of MOLP in comparison to WWF was much higher. Additionally, a 5% MOLP-based value-added bread demonstrated significantly higher antioxidant activities. The overall acceptability scores for WWF-leavened bread decreased progressively as MOLP addition levels increased.	[23]
Gluten-free bread	Fortification of rice semolina gluten-free bread with different amounts of MOLP (2.5%, 5%, 7.5%, and 10%). The TPC and antioxidant activity were determined. A texture and sensory analysis were also performed.	The addition of MOLP resulted in a significant decrease in the volume of the bread samples, except for the 2.5% MOLP. Additionally, a slight decrease in hardness and chewiness was observed with the addition of 2.5% and 10% MOLP. The TPC and antioxidant activity increased as the amount of MOLP increased. Among all MOLP-containing bread samples, the most acceptable bread was the one containing 2.5% MOLP in comparison to the control.	[24]
Biscuits	Production of biscuits with WWF substituted with MOLP (2.5%, 5%, and 10%). Evaluated the nutritional content, bioactive compounds, antioxidant, physical, and α-amylase inhibitory properties. The sensory attributes of the cookies were also determined.	MOLP-supplemented cookies had a significant enhancement in their bioactive compounds, antioxidant, and α-amylase inhibitory properties. Protein, ash, fat, and fibre contents were significantly increased in MOLP-substituted cookies. The sensory acceptance of the cookies decreased with increasing levels of WWF substitution.	[25]
Biscuits	Preparation of biscuits by substituting WWF with 5%, 10%, and 15% MOLP. The effect of MOLP on the rheological, microstructural, nutritional, textural, and organoleptic characteristics of biscuits was tested.	The addition of MOLP led to higher water absorption, softer dough, and an altered cookie texture. Sensory evaluation favoured cookies with 10% MOLP. Nutritional components like protein, iron, calcium, β-carotene, and dietary fibre increased with a higher MOLP content (0–15%).	[26]

MOLP—*Moringa oleifera* leaf powder; TPC—total phenolic content; WWF—whole wheat flour.

**Table 2 antioxidants-12-02069-t002:** List of ingredients for each bread formulation.

Ingredient	NC-Br	MP-Br	ME-Br
Water (g)	210	210	210
Salt (g)	4	4	4
Wheat flour (g)	360	342	352.55
Yeast (g)	2	2	2
MOLP (g)	-	18	-
*M. oleifera* extract (g)	-	-	7.45

MOLP—*Moringa oleifera* leaf powder; NC-BR—Negative control bread; MP-Br—Bread with 5% flour substitution with MOLP; ME-Br—Bread with *M. oleifera* extract flour substitution corresponding to 5% of MOLP.

**Table 3 antioxidants-12-02069-t003:** List of ingredients for each biscuit formulation.

Ingredient	NC-Bi	MP-Bi	ME-Bi
Butter (g)	125	125	125
Sugar (g)	125	125	125
No. Eggs	1	1	1
Wheat flour (g)	250	237.5	244.82
Yeast (g)	2	2	2
MOLP (g)	-	12.5	-
*M. oleifera* extract (g)	-	-	5.18

MOLP—*Moringa oleifera* leaf powder; NC-Bi—Negative control biscuit; MP-Bi—Biscuit with 5% flour substitution with MOLP; ME-Bi—Biscuit with *M. oleifera* extract flour substitution corresponding to 5% of MOLP.

**Table 4 antioxidants-12-02069-t004:** Extraction yields reported in the literature for the extraction of phenolic compounds from *M. oleifera* leaf powder.

Extraction Method	Extraction Conditions	Extraction Yield (%)	Ref.
UAE	Solvent: 70% ethanol *w*/*v* ratio: 1:40 Time: 30 min + 2.5 h Temperature: RT + 50 °C	34.1 ± 0.9	[28]
Sonication	Solvent: 80% ethanol *w*/*v* ratio: 1:50 Time: 30 min (×3)	56.44 ± 0.82	[34]
Maceration	Solvent: 50% or 70% ethanol *w*/*v* ratio: 1:40 Time: 72 h Temperature: RT	Et50: 38.34 ± 1.17 Et70: 40.50 ± 1.24	[35]
Percolation	Solvent: 50% or 70% ethanol	Et50: 34.47 ± 1.41 Et70: 32.75 ± 1.93	
Soxhlet	Solvent: 50% or 70% ethanol *w*/*v* ratio: 1:50 Time: 20 h	Et50: 33.58 ± 1.58 Et70: 35.87 ± 1.12	

UAE—ultrasound-assisted extraction; *w*/*v*—weight of sample/volume of solvent (g/mL); Et50—50% ethanol solvent; Et70—70% ethanol solvent.

**Table 5 antioxidants-12-02069-t005:** Results from the bioactive characterisation of *Moringa oleifera* extract.

TPC (mg_GAE_/g_extract_)	Antioxidant Capacity (IC_50_—mg_extract_/L) (TEAC—mg_TE_/g_extract_)		Antibacterial Capacity (d_halo_—mm)		α-Amylase Inhibition Capacity (%)
138.2 ± 17.0	DPPH	ABTS	*E. coli*	*S. aureus*	94.1 ± 0.4
	544.0 ± 7.9 12.8 ± 0.2	115.2 ± 4.9 32.8 ± 1.4	ND	ND	

ABTS—2,2′-azino-bis(3-ethylbenzothiazoline-6-sulfonic acid); DPPH—2,2-diphenyl-1-picrylhydrazyl; GAE—gallic acid equivalents; IC_50_—concentration of extract needed to inhibit 50% of the free radicals; ND—not detected; TE—Trolox equivalents; TEAC—Trolox equivalent antioxidant capacity; TPC—total phenolic content. The results are expressed as means ± standard deviations of three independent measurements.

**Table 6 antioxidants-12-02069-t006:** Main phenolic compounds present in the *M. oleifera* extract quantified by HPLC-DAD.

Compounds	RT (min)	Calibration Curves	R^2^	IDL (mg/L)	IQL (mg/L)	Phenolic Concentration (mg_compound_/g_extract_)
Catechin	24.38	A = 1.57 × 10^5^ C − 7.93 × 10^5^	0.9861	41.50	138.34	1.24
Epicatechin	30.34	A = 4.15 × 10^5^ C − 1.56 × 10^6^	0.9983	5.82	19.39	0.09
Caffeic acid	29.23	A = 5.56 × 10^5^ C − 1.56 × 10^6^	0.9992	4.03	13.43	0.07
Chlorogenic acid	26.62	A = 1.91 × 10^5^ C − 1.16 × 10^5^	0.9999	2.84	9.48	0.01
Quercetin	52.79	A = 7.37 × 10^5^ C − 2.68 × 10^5^	0.9994	1.37	4.58	0.01

A—peak area; C—standard concentration (mg/L); IDL—instrumental detection limit; IQL—instrumental quantification limit; R^2^—coefficient of determination; RT—retention time.

**Table 7 antioxidants-12-02069-t007:** Results from microbial contamination analysis of breads and biscuits during the study period.

Formulation	Timepoint	LSA	RBC
		(CFU/mL)	
NC-Br	t0	ND	ND
	t1	>30 × 10^5^	>30 × 10^5^
MP-Br	t0	ND	ND
	t1	>30 × 10^5^	ND
ME-Br	t0	ND	ND
	t1	3.05 × 10^5^	ND
NC-Bi	t0	ND	ND
	t1	ND	ND
	t2	ND	ND
	t3	ND	ND
MP-Bi	t0	ND	ND
	t1	ND	ND
	t2	ND	ND
	t3	ND	ND
ME-Bi	t0	ND	ND
	t1	ND	ND
	t2	ND	ND
	t3	ND	ND

ND—not detected; LSA—Lauryl Sulfate Agar medium; RBC—Rose-Bengal Chloramphenicol Agar medium. NC-Br—bread with no additives (negative control); MP-Br—bread with 5% MOLP; ME-Br—bread with extract corresponding to 5% MOLP. NC-Bi—biscuits with no additives (negative control); MP-Bi—biscuits with 5% MOLP; ME-Bi—biscuits with extract corresponding to 5% MOLP. The analysis was performed at different timepoints: for bread, t0—day after production and t1—four days after production; for biscuits, t0—day after production, t1—two weeks after production, t2—four weeks after production, and t3—seven weeks after production.

## Data Availability

The data presented in this study are available on request from the corresponding author.

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
