# Peer review of "Elevating Cereal-Based Nutrition: Moringa oleifera Supplemented Bread and Biscuits"

_antioxidants, 2023, doi:10.3390/antiox12122069_

Round 1

Reviewer 1 Report

Comments and Suggestions for Authors

The study explored the possibility to produce fortified bread and biscuits supplemented with Moringa oleifera leaf powder and its phenolic-rich extract. The paper is reporting a broad overview, from chemical extraction and characterization to the sensorial acceptability of cereal-based products, also considering the a-amylase activity and cytotoxicity. However, it requires an accurate revision, mainly regarding the section of Introduction that is like a review article and of the Results and Discussions.

Below some recommendations.

Introduction is too long. It takes up 4 and a half pages.

Line 50. It would be better to use the expression ‘plant nutraceuticals’ instead of ‘nutraceutical plant’.

Line 51. Why the authors used the expression ‘food additive’ and not ‘ingredient’?

Lines 62-63. The information reported in these lines are redundant, as just mentioned in lines 50-52. The authors could integrate the information.

Lines 79-80. The reference 14 is not related to food security. Check.

Line 99. What the authors do intend for ‘food insecurity’? Perhaps, lack of food availability? See also line 646.

Lines 101-146. Since this work does not have the purpose of in vivo medical studies, I would try more than anything to highlight the purpose of this research and the innovations in terms of knowledge compared to the literature.

Table 1 is more suitable for a review than a research article. So, the authors should reduce it by reporting only the most significant information (standardizing the measure units) or, better yet, remove it and report the most significant aspects directly in the text before the purpose of the work.

Lines 184-189. No citation of a reference method. The same for lines 242-254, lines 293-299.

Line 267. What type of oven were the biscuits baked in?

Line 300. Did the authors make a consumer test or a panel test by trained personnel?

Line 308. The authors should add that the data have been presented as mean ± standard deviation.

Lines 315-317. It is not necessary to specify the extraction method and the choice of solvent as it is already reported in the materials and methods section.

Lines 318-319. The extraction yield should be moved to the materials and methods section.

Lines 338-341. Please, check carefully.

Lines 342-362. Revise this part in comparing the results more concisely: supporting previous findings or contradicting previous findings, without describing extraction conditions or reporting too many data of other authors as there are the references.

Lines 363-364. The authors refer to positive control or samples. The unique data reported in Table 4 are the characterization of Moringa oleifera extract. Please, check.

Lines 373-380. Revise this period and write more concisely. Moreover, although the authors argued about alpha-amylase and not alpha-glucosidase, they comment on both enzymes. Check.

Lines 386-388. In the introduction the authors reported the most common PCs (lines 69-71) quite different from the results here reported, except for caffeic acid.

Lines 413-417. Uniform the safety level of extract with the introduction (see line 106) and conclusions (line 636). Is it necessary to report the information in brackets? The reference to resazurin assay is unnecessary as just reported in the materials and methods section.

Lines 549-553. The information reported in these lines are redundant as just found in materials and method section and in the legend of Table 6.

Line 566. After ‘room temperature’ add ‘(t1)’.

Lines 569-576. As MP-Br data was unexpected respect ME-Br, what hypothesis do the authors make?

Line 580. After ‘NC-Br’ add ‘(t1)’.

Lines 592-595. This introduction can be deleted as just reported in materials and methods section. Moreover, lines 595-597 should be moved and integrated with materials and methods section.

Figure 7. No statistics are reported in the Figure. Change ‘general acceptability’ with ‘overall acceptability’ in accordance with the text.

Comments on the Quality of English Language

Minor editing of English language required

Author Response

Journal: Antioxidants

Manuscript ID: antioxidants-2715830

Note: The authors wish to express their appreciation to the Reviewers for their valuable comments on the manuscript. We hope that our revision can answer the queries posed and may reflect an effective improvement of our work.

Author's Reply to the Review Report (Reviewer1)

Comments and Suggestions for Authors

The study explored the possibility to produce fortified bread and biscuits supplemented with Moringa oleifera leaf powder and its phenolic-rich extract. The paper is reporting a broad overview, from chemical extraction and characterization to the sensorial acceptability of cereal-based products, also considering the a-amylase activity and cytotoxicity. However, it requires an accurate revision, mainly regarding the section of Introduction that is like a review article and of the Results and Discussions.

Below some recommendations.

NOTE: The authors are grateful for the comment. The line numbers indicated in the answers represent the number of the line of the manuscript where the modification was performed when the modifications are visible.

Introduction is too long. It takes up 4 and a half pages.

Answer: The authors agree with the comment and shortened the introduction.

Line 50. It would be better to use the expression ‘plant nutraceuticals’ instead of ‘nutraceutical plant’.

Answer: Thank you for the suggestion. The authors substituted ´nutraceutical plant´ for ´plant nutraceutical´ (line 50).

Line 51. Why the authors used the expression ‘food additive’ and not ‘ingredient’?

Answer: Thank you for the comment. In fact, the expression ‘ingredient‘ is more adequate in this context, therefore ‘food additive’ was substituted for ‘ingredient’ (line 51).

Lines 62-63. The information reported in these lines are redundant, as just mentioned in lines 50-52. The authors could integrate the information.

Answer: Thank you for the comment. The repeated information was removed.

Lines 79-80. The reference 14 is not related to food security. Check.

Answer: Thank you for your comment. The food security statement was incorrectly placed before reference number 14 which is related to the previous statement about antioxidants. Therefore, the food security statement was moved to the front of reference 14 (lines 79-81). 

Line 99. What the authors do intend for ‘food insecurity’? Perhaps, lack of food availability? See also line 646.

Answer: The authors intend food security as the condition where individuals or households lack consistent access to enough, safe, nutritious foods to meet their dietary needs for an active and healthy life, while lack of food availability specifically focuses on the physical presence and accessibility of food in a given region or community. The authors therefore thought it was more appropriate to use ‘food insecurity’ instead of ‘lack of food availability’ in this context. The authors added a sentence to the manuscript to clarify the concept of food insecurity (lines 100-101).

Lines 101-146. Since this work does not have the purpose of in vivo medical studies, I would try more than anything to highlight the purpose of this research and the innovations in terms of knowledge compared to the literature.

Answer: Thank you for your comment. The authors did not understand this observation. In the introduction, the authors already specify the purpose of this research and the innovations in terms of knowledge compared to the literature (lines 145-158).

Table 1 is more suitable for a review than a research article. So, the authors should reduce it by reporting only the most significant information (standardizing the measure units) or, better yet, remove it and report the most significant aspects directly in the text before the purpose of the work.

Answer: Thank you for your comment. Although the authors understand the comment, the other reviewers made no observations regarding this aspect and therefore, the authors preferred to keep the table with only the most significant information in order to have the studies presented in a neater and more straightforward way instead of including them directly in the text.

Lines 184-189. No citation of a reference method. The same for lines 242-254, lines 293-299.

Answer: Thank you for the comment. The citations were added, as suggested.

Line 267. What type of oven were the biscuits baked in?

Answer: The biscuits were baked in a convection oven from the brand Gaggenau from Germany. This information was added to the Materials and Methods section (line 301).

Line 300. Did the authors make a consumer test or a panel test by trained personnel?

Answer: The consumer’s test was not performed by a trained personnel. That information was added to the manuscript (line 342).

Line 308. The authors should add that the data have been presented as mean ± standard deviation.

Answer: Thank you for the comment. The information was added, as suggested (lines 353-354).

Lines 315-317. It is not necessary to specify the extraction method and the choice of solvent as it is already reported in the materials and methods section.

Answer: Thank you for the comment. The authors believed that it is important to discuss the rationale for the choice of the method and solvent. Therefore, and considering the observations of the other reviewers, the authors opted to maintain this information in this section and add a comparison with alternative extraction methods, considering factors like efficiency and environmental impact (lines 357-379).

Lines 318-319. The extraction yield should be moved to the materials and methods section.

Answer: Thank you for the comment. Considering the observations of the other reviewers, the authors added a comparison of the extraction yield with other literature reports and, therefore, decided to maintain this information in the Results section.

Lines 338-341. Please, check carefully.

Answer: The authors checked the text present in the mentioned lines and did not find any mistakes.  

Lines 342-362. Revise this part in comparing the results more concisely: supporting previous findings or contradicting previous findings, without describing extraction conditions or reporting too many data of other authors as there are the references.

Answer: Thank you for the comment. The authors revised this section and removed some information to make the information more concise, as suggested.

Lines 363-364. The authors refer to positive control or samples. The unique data reported in Table 4 are the characterization of Moringa oleifera extract. Please, check.

Answer: Thank you for the comment. The authors substituted ‘positive control or samples’ for ‘M. oleifera leaf extract sample’ (line 434).

Lines 373-380. Revise this period and write more concisely. Moreover, although the authors argued about alpha-amylase and not alpha-glucosidase, they comment on both enzymes. Check.

Answer: The authors agree with the comment and removed the sentence that accessed the results from another study regarding the enzyme α-glucosidase since it is not relevant to the present study (lines 454-456).

Lines 386-388. In the introduction the authors reported the most common PCs (lines 69-71) quite different from the results here reported, except for caffeic acid.

Answer: Thank you for your comment. The phenolic compounds identified by HPLC in the present work are also very commonly reported in the literature. This information was added to the Introduction section in order not to mislead the readers (lines 70-71).

Lines 413-417. Uniform the safety level of extract with the introduction (see line 106) and conclusions (line 636). Is it necessary to report the information in brackets? The reference to resazurin assay is unnecessary as just reported in the materials and methods section.

Answer: Thank you for the comment. The authors altered the information in brackets and removed the reference to the resazurin assay, as suggested.

Lines 549-553. The information reported in these lines are redundant as just found in materials and method section and in the legend of Table 6.

Answer: The authors agreed with the comment and eliminated that information.

Line 566. After ‘room temperature’ add ‘(t1)’.

Answer: Thank you for the comment. The information was added, as suggested (line 659).

Lines 569-576. As MP-Br data was unexpected respect ME-Br, what hypothesis do the authors make?

Answer: The authors hypothesize that the reason for this unexpected result might have been the fact that although the bread incorporated with MOLP (MP-Br) contained a high total phenolic content (TPC) value (higher than the bread containing MOLP extract, ME-Br), the extract might contain other compounds that have antimicrobial effect (such as alkaloids and glycosides) in a higher concentration which contribute to the disparity in results between MP-Br and ME-Br. The authors think this observation is important and added this information to the article (lines 671-674). 

Line 580. After ‘NC-Br’ add ‘(t1)’.

Answer: Thank you for the comment. The information was added, as suggested (line 677).

Lines 592-595. This introduction can be deleted as just reported in materials and methods section. Moreover, lines 595-597 should be moved and integrated with materials and methods section.

Answer: Thank you for the comment. The authors performed the suggested alterations.

Figure 7. No statistics are reported in the Figure. Change ‘general acceptability’ with ‘overall acceptability’ in accordance with the text.

Answer: The authors substituted ‘general acceptability’ for ‘overall acceptability’ in Figure 7 and added the statistical analysis to the figure, as suggested.

Reviewer 2 Report

Comments and Suggestions for Authors

Author Response

Author's Reply to the Review Report (Reviewer 2)

Comments and Suggestions for Authors

The study addresses a crucial aspect of improving the nutritional content of staple foods, such as bread and biscuits, which are widely consumed globally. Fortifying these foods with Moringa oleifera leaf powder (MOLP) and its phenolic-rich extract is a commendable approach to tackle malnutrition. The phenolic extract from MOLP demonstrated significant antioxidant properties, as evidenced by low IC50 values in ABTS and DPPH assays. Additionally, the extract exhibited antibacterial properties and inhibition of α-amylase, making it a potential candidate for addressing oxidative stress, bacterial contamination, and diabetes. The study identifies specific phenolic compounds, such as catechin, epicatechin, and caffeic acid, contributing to the health-promoting properties of MOLP. This adds depth to the understanding of the bioactive components responsible for the observed effects.

In addition, the incorporation of MOLP and its extract into bread and biscuits resulted in fortified functional foods with improved total phenolic content, antioxidant activity, and the ability to inhibit microbial growth. This not only enhances the nutritional quality but also extends the shelf-life of the products, which is a practical benefit.

The sensory analysis revealing a preference for products incorporated with the extract over those with MOLP suggests that the fortified products not only offer health benefits but are also well received in terms of taste and overall sensory experience.

The study highlights the use of MOLP and its phenolic-rich extract as an environmentally sustainable strategy. This is important for promoting sustainable and eco-friendly practices in the food industry.

NOTE: The authors are grateful for the comment. The line numbers indicated in the answers represent the number of the line of the manuscript where the modification was performed when the modifications are visible.

Areas for Consideration:

Extraction Method: While the study mentions the Soxhlet extraction method for obtaining the phenolic extract, it would be beneficial to discuss the rationale for choosing this method and compare it with alternative extraction methods, considering factors like efficiency and environmental impact.

Answer: The authors added to the manuscript the explanation of why this extraction method was chosen and compared the extraction yields and environmental impact of this extraction method with other methods (lines 357-379).

Cytotoxicity and Safety: The study mentions no cytotoxicity towards human fibroblasts, which is positive. However, providing more detailed information on safety assessments, potential allergens, or any other safety considerations related to the consumption of the fortified products would enhance the comprehensiveness of the study.

Answer: Thank you for the comment. The focus of the present work was to analyse the biological properties of the Moringa oleifera extract as well as to assess the possibility of incorporating it in cereal-based products. The authors understand that is of utmost importance to ensure the safety of the consumers and, therefore, performed a cytotoxicity analysis towards human fibroblasts. Although the authors did not perform other safety assessments, since this work intended to be a proof of concept of the possibility of incorporating the M. oleifera powder and extract into bread and biscuits, the authors understand their importance and will take this into consideration in future work. A paragraph about future work was added to the Conclusion section (lines 751-759).

Material and methods: there is a lack of explanation Shelf-Life, for how long did you store the samples? How did you storage the samples? Temperature, humidity, time?? Why didn’t you analyse colour by colorimeter?

Answer: Thank you for the comment. The authors stored the samples during the time the analyses were performed: one week for bread and seven weeks for biscuits. The bread was only stored for one week due to its normal shelf-life of about 5 days; the biscuits have extended shelf-life and, therefore, were stored for a prolonged period of time. Both products were stored in plastic bags, protected from the light, at room temperature. The information about the storage of the samples was added to the manuscript (lines 302-303). Regarding the characterization of the samples, the authors opted to focus on the antioxidant properties of the products rather than their physical characteristics (such as colour, texture, …), which were only briefly analysed in the sensory analysis. However, the authors understand the importance of evaluating the effect of the addition of M. oleifera on the physical parameters of the products and will take this into consideration in future work. A paragraph about future work was added to the Conclusion section (lines 751-759).

The panel for the sensory analysis is trained? Are consumers? If it is a consumers panel 20 taster are very few number.. please specify

Answer: Thank you for the comment. The information about the panellists was added to the manuscript, as suggested (line 342).

From my point of view, when we observe the bread with the highest percentage of Moringa, it is a not well structure bread. I don’t think this bread was correctly formulated and therefore, the results obtained are questionable.

Answer: Thank you for the comment. The authors believe that all formulations were well formulated. When you refer to the bread with the highest percentage of M. oleifera do you mean the bread with the powder (Figure 3B)? If so, the changes in the structure of the bread containing the powder (which was denser, shorter, and less voluminous compared to the regular bread) were also reported by other authors and may be explained by the powder's binding properties and moisture retention. This information is already reported in the manuscript (lines 543-549).

In conclusion, the study presents a promising strategy for fortifying commonly consumed staple foods with MOLP and its phenolic-rich extract. The demonstrated health benefits, sensory acceptance, and environmental sustainability aspects make it a compelling approach for addressing malnutrition and improving the quality of cereal-based products.

However, it is not well structured. The manuscript should be rewritten and the bread should be reformulated

Answer: Thank you for the comment. The authors performed several alterations to the manuscript to try to improve it.

Reviewer 3 Report

Comments and Suggestions for Authors

The paper presents the effect of MOLP and MOLP extracts in bread and biscuits. Generally, the paper is well-structured, easy to read and complex. Some remarks are made below.

Abstract: it is well constructed.

Introduction: This section presents enough background to support the study. Furthermore, the aim of the paper is well presented.

Methods: Why 5% addition level?

L303: Please mention if the panelists were trained or not. Also, what have the authors used for mouth cleaning between samples?

The number of replicates for each analysis must be mentioned.

Results and discussion:

L376-380: References are needed when you give numbers.

L399 Which study? reference is needed.

L592-597 I suggest removing this part because it is presented in the methods.

Figure 7: Statistics is missing.

L 616 Reference is needed.

The authors mention some health benefits in the manuscript without any determination to demonstrate this (for example the predicted glycemic index or other analysis in vitro to demonstrate the health benefits). Thus, the authors have 2 options: reduce the affirmations related to health benefits (you can keep them in the Introduction), or perform digestibility, PGI or other analysis to support the affirmations. 

Author Response

Author's Reply to the Review Report (Reviewer 3)

Comments and Suggestions for Authors

The paper presents the effect of MOLP and MOLP extracts in bread and biscuits. Generally, the paper is well-structured, easy to read and complex. Some remarks are made below.

NOTE: The authors are grateful for the comment. The line numbers indicated in the answers represent the number of the line of the manuscript where the modification was performed when the modifications are visible.

Abstract: it is well constructed.

Answer: The authors are grateful for the comment.

Introduction: This section presents enough background to support the study. Furthermore, the aim of the paper is well presented.

Answer: The authors are grateful for the comment.

Methods: Why 5% addition level?

Answer: In the literature, the most common addition levels studied are, in the case of bread, 5-15%. In the case of biscuits, the addition levels described in the literature can be slightly higher, but to compare both products, the same substitution level was applied in both foods. The authors opted for the 5% addition level due to the high amounts of extract necessary for higher addition levels. Nevertheless, the results with these substitution levels were already quite promising and the authors intend to study higher levels in future work, to understand what is the technological limit for the supplementation level, as well as the effect on the sensory properties. A brief sentence was added to the Conclusion section about future work (lines 744-746).

L303: Please mention if the panelists were trained or not. Also, what have the authors used for mouth cleaning between samples?

Answer: Thank you for the comment. The information about the panellists was added to the manuscript, as suggested (line 342). For mouth cleaning between samples, the authors provided plain crackers to the participants.

The number of replicates for each analysis must be mentioned.

Answer: Thank you for the comment. The information about the replicates of each analysis was added to the manuscript in the Materials and Methods section, as suggested.

Results and discussion:

L376-380: References are needed when you give numbers.

Answer: Thank you for the comment. The authors added the reference of the study.

L399 Which study? reference is needed.

Answer: The authors added the reference.

L592-597 I suggest removing this part because it is presented in the methods.

Answer: Thank you for the comment. The information was removed, as suggested.

Figure 7: Statistics is missing.

Answer: The authors added the statistical analysis to Figure 7.

L 616 Reference is needed.

Answer: Thank you for the comment. The authors added a reference.

The authors mention some health benefits in the manuscript without any determination to demonstrate this (for example the predicted glycemic index or other analysis in vitro to demonstrate the health benefits). Thus, the authors have 2 options: reduce the affirmations related to health benefits (you can keep them in the Introduction), or perform digestibility, PGI or other analysis to support the affirmations.

Answer: Thank you for the comment. The authors agree with the comment and therefore reduced the affirmations related to health benefits in the manuscript.

Round 2

Reviewer 1 Report

Comments and Suggestions for Authors

The paper is ready for publication

Reviewer 2 Report

Comments and Suggestions for Authors

Thank you for your review.

Reviewer 3 Report

Comments and Suggestions for Authors

The paper was improved according to the suggestions.

However, 20 untrained panelists for the sensory analysis are too few. A number of 80 would be suitable for this type of consumer.